

# Cryoprotective activities of FK20, a human genome-derived intrinsically disordered peptide against cryosensitive enzymes without a stereospecific molecular interaction

Naoki Matsuo[1,*], Natsuko Goda[1,*], Takeshi Tenno[1,2] and Hidekazu Hiroaki[1,2]

[1] Graduate School of Pharmaceutical Sciences, Laboratory of Structural and Molecular Pharmacology, Nagoya University, Nagoya, AICHI, JAPAN
[2] BeCellBar, LLC., Nagoya, Aichi, Japan
* These authors contributed equally to this work.

Corresponding author
Hidekazu Hiroaki,
hiroaki.hidekazu@f.mbox.
nagoya-u.ac.jp

## ABSTRACT

**Background:** Intrinsically disordered proteins (IDPs) have been shown to exhibit cryoprotective activity toward other cellular enzymes without any obvious conserved sequence motifs. This study investigated relationships between the physical properties of several human genome-derived IDPs and their cryoprotective activities.
**Methods:** Cryoprotective activity of three human-genome derived IDPs and their truncated peptides toward lactate dehydrogenase (LDH) and glutathione S-transferase (GST) was examined. After the shortest cryoprotective peptide was defined (named FK20), cryoprotective activity of all-D-enantiomeric isoform of FK20 (FK20-D) as well as a racemic mixture of FK20 and FK20-D was examined. In order to examine the lack of increase of thermal stability of the target enzyme, the CD spectra of GST and LDH in the presence of a racemic mixture of FK20 and FK20-D at varying temperatures were measured and used to estimate $T_m$.
**Results:** Cryoprotective activity of IDPs longer than 20 amino acids was nearly independent of the amino acid length. The shortest IDP-derived 20 amino acid length peptide with sufficient cryoprotective activity was developed from a series of TNFRSF11B fragments (named FK20). FK20, FK20-D, and an equimolar mixture of FK20 and FK20-D also showed similar cryoprotective activity toward LDH and GST. $T_m$ of GST in the presence and absence of an equimolar mixture of FK20 and FK20-D are similar, suggesting that IDPs' cryoprotection mechanism seems partly from a molecular shielding effect rather than a direct interaction with the target enzymes.

## INTRODUCTION

Intrinsically disordered proteins (IDPs) are an important class of proteins, that is widely associated with broad biological processes (*Wright & Dyson, 1999*; *Romero et al., 2001*; *Dunker et al., 2008*; *Tompa & Fersht, 2009*; *Uversky & Dunker, 2010*). The most unique feature of IDPs is that they lack stable and compact tertiary structures alone under physiological conditions, while some of them eventually fold into stable structures during specific interaction. Currently, at least two major biological functions of IDPs have been proposed: "coupling folding and binding" (*Dyson & Wright, 2002*; *Sugase, Dyson & Wright, 2007*; *Higo, Nishimura & Nakamura, 2011*) and "accelerated association with partner molecules by the fly-casting mechanism" (*Levy, Onuchic & Wolynes, 2007*; *Sugase, Dyson & Wright, 2007*; *Chen, 2009*). The former function is amino acid sequence dependent, because at least some part of the IDP must adopt into a fixed conformation with specific molecular contacts upon target binding. The latter function is assumed as less sequence dependent; a higher flexibility in the extended conformation of the IDP region that interconnects two functional domains, is seemingly more important. Recently, we and other researchers proposed a third physiological function of IDPs as protectants from environmental stresses, such as freezing and desiccation (*Hughes & Graether, 2011*; *Hughes et al., 2013*; *Boothby et al., 2017*; *Matsuo et al., 2018*). In this study, we focused on the cryoprotective activity of IDPs.

Plant dehydrins (DHNs) are extensively well studied examples of cryoprotective IDPs (*Allagulova et al., 2003*; *Hanin et al., 2011*; *Graether & Boddington, 2014*). DHNs belong to a family of late embryogenesis abundant (LEA) proteins that are major contributors to the development of desiccation tolerance during plant seed maturation (*Goyal, Walton & Tunnicliffe, 2005*; *Hincha & Thalhammer, 2012*; *Amara et al., 2014*). DHNs are characterized by the presence of one or more uniquely conserved sequence motifs, Lys-rich (K-), Tyr-rich (Y-), and Ser-rich (S-) segments (*Hughes & Graether, 2011*). Many *in vitro* studies have demonstrated that DHNs effectively prevent inactivation of the model reporter enzyme, lactate dehydrogenase (LDH), during repeated freeze/thaw cycles (*Hughes & Graether, 2011*; *Cuevas-Velazquez, Rendon-Luna & Covarrubias, 2014*). A possible mechanism underlying DHN cryoprotection involves "molecular shields" that prevent stochastic direct contacts between enzyme molecules (*Chakrabortee et al., 2012*; *Hughes et al., 2013*). Two independent research groups found that the cryoprotective activity of DHNs and their artificial variants roughly correlated with their hydrodynamic radius ($R_H$) rather than their amino acid sequences (*Hughes et al., 2013*; *Cuevas-Velazquez, Rendon-Luna & Covarrubias, 2014*; *Ferreira et al., 2018*), partly supporting the molecular shielding hypothesis.

Strongly encouraged by Graether and colleagues' work (*Hughes & Graether, 2011*; *Hughes et al., 2013*; *Ferreira et al., 2018*), we have uncovered evidence supporting the molecular shielding hypothesis of cryoprotective IDPs through a different approach (*Matsuo et al., 2018*). If the cryoprotective action is driven solely by the molecular shielding effect, any other IDPs could also exert cryoprotective activity. Indeed, we found a potential

example in the literature, silk worm sericin, a Ser-rich IDP (*Tsujimoto et al., 2001*). We expanded the cryoprotective IDP concept to include other evolutionarily unrelated IDPs, such as human genome-derived IDPs of 36 to 44 amino acid residues, rather than DHNs or LEAs. In our previous study (*Matsuo et al., 2018*), we demonstrated that all the examined IDPs derived from the human genome exerted cryoprotective activity toward not only the model enzyme LDH, but also glutathione-S transferase (GST) and green fluorescent protein (GFP). In detail, 53 candidate IDP genes from human genome were first predicted by a bioinformatics method, and then, among them, 35 IDP peptides systematically proven as the flexible disordered peptide segments by the NMR-based indirect IDP-assessment methods developed by us (*Goda et al., 2015b*). Accordingly, the five randomly-selected human genome-derived IDP peptides among the 35 peptides were demonstrated to have substantial cryoprotective activities against LDH, GST, and GFP (*Matsuo et al., 2018*). However, in that study, all five IDPs showed similar levels of cryoprotective activity; we could not identify further sequence-activity relationships in the human genome-derived IDPs.

In this study, we selected three IDPs, C1, D10, and E1, for further analysis. We succeeded in minimizing the length of the human-genome-derived IDPs with practical cryoprotective activity. We found that FK20—the 20 amino acid fragment (position 24-43) from the tumor necrosis factor receptor superfamily member 11B (TNFRS11B) precursor—showed substantial cryoprotective activity toward LDH. Then we compared the cryoprotective activity of FK20 and its all-D-enantiomeric isomer and found that the cryoprotective activity of FK20 is independent to its chirality. Accordingly, we employed a new technique to use the racemic mixture of FK20 and FK20-D, that do not show CD signal, to investigate the absence of specific molecular effect toward the GST and LDH reporter enzyme by CD spectroscopy. These results suggest a potential use of human genome-derived IDP as a cryopreserving agent for cryosensitive enzymes and proteins.

## MATERIALS & METHODS

### Expression and preparation of the IDP samples

Human genome-derived IDP samples and their truncation mutants were prepared by an *E. coli* expression system optimized for preparing IDP samples, as previously described (*Goda et al., 2015a*). In brief, the N-terminal autoprotease N(pro) from bovine viral diarrhea virus was selected as a fusion partner for protein expression using the pET-based N(pro) fusion protein expression system (*Achmüller et al., 2007*). The IDPs were expressed, N-terminal tags were removed, and finally purified by reversed phase HPLC (COSMOSIL® 5C4-AR-300, Nacalai Tesque, $\phi$4.6 mm × 250 mm) with 0.1% trifluoroacetic acid–acetonitrile solvent system. All the peptides were quantified by UV absorbance at 280 nm, lyophilized, and stored at −30 °C until use. FK20, a 20 amino acid peptide, and its all-D-enantiomeric isomer, FK20-D, were chemically synthesized (Biologica Co. Ltd. Nagoya, Aichi, Japan) with at least 80% purity, and purified by reversed phase HPLC.

## Cryoprotection assay for LDH

Rabbit muscle lactate dehydrogenase (LDH) (L-2500; Sigma-Aldrich, St. Louis, MO, USA) was selected as the reporter enzyme for cryoprotection activity of shorter IDP peptides, using a slightly modified protocol based on Hughes and Graether (*Hughes & Graether, 2011*). The initial LDH solution contained 50 µg/ml in 10 mM sodium phosphate (pH 7.4). In a 1.5-ml microfuge tube, we mixed a 10-µl aliquot of the LDH solution with 10 µl of solution containing each of the individual protectants (IDP peptides) at concentrations ranging from 2.5 to 500 µg/ml. Five cycles of freezing in liquid nitrogen for 30 s and thawing in a water bath at 4 °C for 5 min were applied to each sample. Subsequently, LDH activity was measured using a standard NADH oxidase coupled-enzyme system according to the our previous report (*Matsuo et al., 2018*). For the analysis, we set the LDH activity of the untreated sample (the enzyme without freeze and thaw processes and without the addition of a cryoprotectant) as 100%. All measurements were performed in triplicate.

## Cryoprotection assay for GSH

Glutathione S-transferase from *Schistosoma japonicum* (GST) was selected as the second reporter enzyme for cryoprotection activity of shorter IDP peptides because this enzyme was easily prepared in the laboratory and also suited for the subsequent CD spectroscopy experiments. The cryoprotection assay was performed according to our previous report (*Matsuo et al., 2018*). After five freeze/thaw cycles, GST activity was measured using a standard 1-chloro-2,4-dinitrobenzene (CDNB, 138630; Sigma-Aldrich, St. Louis, MO, USA) assay with a 96-well plate and 2300 EnSpire Microplate Reader (Perkin Elmer, Waltham, MA, USA). For the analysis, we set the GST activity of the untreated sample (the enzyme without freeze and thaw processes and without the addition of a cryoprotectant) as 100%. All measurements were performed three times and the standard deviations were calculated.

## Circular dichroism (CD) measurements

CD spectra between 190 and 300 nm were collected on a J-805 spectropolarimeter (JASCO, Tokyo, Japan). The time constant, scan speed, bandwidth/resolution, and sensitivity of the spectropolarimeter were set at 1 s, 100 nm/min, 1 nm, and 100 mdeg, respectively. We measured a 300 µl solution of 250 µg/ml of FK20, FK20-D or 125 µg/ml each of the FK20/FK20-D mixture with a 10 mM sodium phosphate buffer (pH 7.4) in a quartz cuvette with a 1 mm light path length at 20 °C. Accordingly, we measured CD spectra of a 300 µl solution of 100 µg/ml of GST with 0, 25, 50 and 100 µg/ml each of FK20/FK20-D mixture in the same buffer. Similarly, we measured CD spectra of a 300 µl solution of 100 µg/ml of LDH with 0, 25, 50 and 100 µg/ml each of FK20/FK20-D mixture in the same buffer.

The GST and LDH denaturation temperatures ($T_m$) were determined by CD spectra at varying temperatures from 25 °C to 75 °C or 85 °C by monitoring the ellipticity at 222 nm at the speed of heating by 1 °C/min. Every 10 °C, CD spectra between 200 and 300 nm were automatically collected. For LDH, CD spectra of LDH were measured at 25 °C

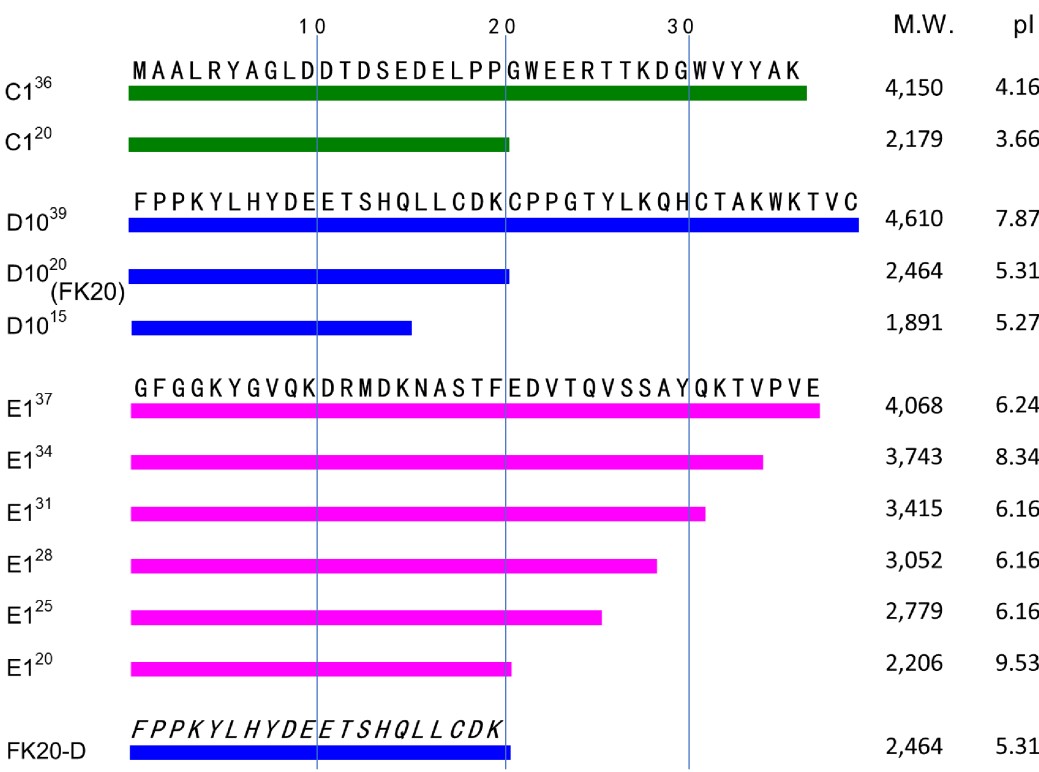

**Figure 1 Amino acid sequences and schematic diagrams of human genome-derived IDPs and their deletion mutants.** Molecular weight (M.W.) and calculated isoelectric point (pI) are also shown. $D10^{20}$ was renamed FK20. Residues of D-amino acids are shown in *italics*. The Ref_seq accession codes, protein names, and the corresponding residue numbers of the human genome-derived IDPs are as follows; C1 (NP_570859.1 *obsolete*, WW domain-containing oxidoreductase isoform 3, 1-36), D10 (NP_002537.3, tumor necrosis factor receptor superfamily member 11B precursor, 24-62), and E1 (NP_005222.2, src substrate cortactin isoform a, 305-342).

and 85 °C. A moving average of CD values of each five temperature points were plotted, and three straight lines corresponding to the baseline, the plateau, and the slope, were indicated. $T_m$ was determined as the temperature of 50% denatured state.

## RESULTS

### Cryoprotective activity of shorter IDPs toward LDH and GST

To determine the minimal length of IDPs showing a substantial cryoprotective activity toward LDH and GST, we constructed several recombinant plasmids containing coding regions of the human genome-derived IDPs C1, D10, E1, and a series of their C-terminal truncated mutants. Amino acid sequences with schematic diagrams of IDPs and their truncated mutants are shown in Fig. 1. Human genome-derived IDPs C1, D10, and E1 consisted of 36, 37 and 39 amino acids, respectively. These peptides were proven as IDPs by both CD spectra and $^1$H-$^{15}$N 2D-NMR spectra in our previous studies (*Goda et al., 2015b*; *Matsuo et al., 2018*). We used the PONDR server (http://pondr.com) (*Obradovic et al., 2005*) to predict whether the series of C-terminal truncated peptides were also disordered. Accordingly, the charge-hydrophaty plots (Uversky plots) were shown in

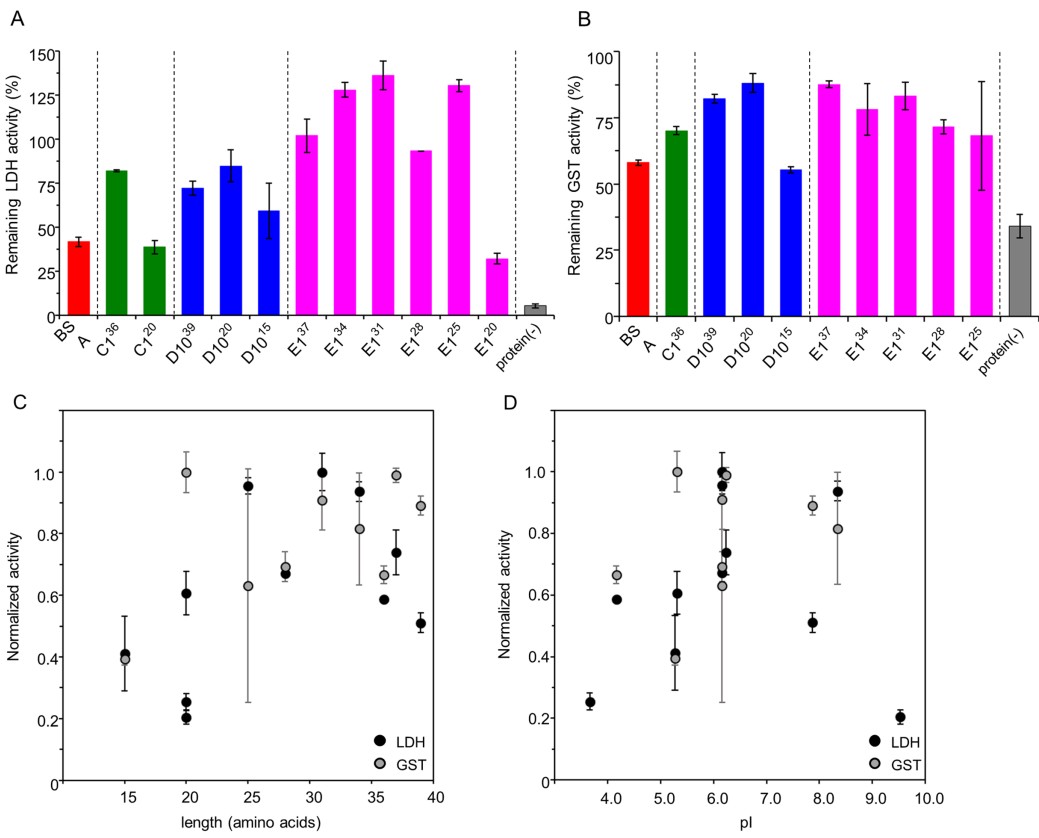

**Figure 2 Cryoprotective activity of human genome derived IDPs against model enzymes LDH and GST.** Cryoprotective activity of human genome-derived IDPs and control proteins towards LDH (A) and GST (B). Dependency of the normalized cryoprotective activity against amino acid lengths (C) and calculated isoelectric points, pI (D). (A) LDH activities (final concentration of 50 µg/ml) after freeze-thawing in the presence of 50 µg/ml indicated additive IDP peptides. The LDH activity of the untreated sample was set to 100%. Protein(-) indicates LDH activity without cryoprotectant. The error bars indicate standard deviation. (B) GST activities (final concentration of 280 µg/ml) after freeze-thawing in the presence of 50 µg/ml indicated additive IDP peptides. The GST activity of the untreated sample was set to 100%. IDPs and peptides are colored black. Protein(-) indicates GST activity without cryoprotectant. The error bars indicate standard deviation. (C) Cryoprotective activities of each cryoprotectants towards LDH and GST are normalized and plotted against their amino acid length. (D) Normalized cryoprotective activities are plotted against pI of each cryoprotectants.

Fig. S1. Cryoprotective activity of all peptides toward LDH at concentrations of 5, 10, 25, 50 and 500 µg/ml were measured. All peptides showed stronger cryoprotective activity than BSA in a concentration-dependent manner, with none showing less than 90% cryoprotection at the highest concentration (500 µg/ml) (data not shown). We compared these cryoprotective effects at the intermediate concentration (50 µg/ml) (Fig. 2A) and found that the cryoprotective activity of most of IDPs was independent of their amino acid lengths, except the three shorter peptides, C1[20], D10[15] and E1[20]. These peptides showed decreased cryoprotection activity of less than 60%. As a result, D10[20] was the shortest peptide with practical cryoprotective activity toward LDH at 50 µg/ml.

Subsequently, we examined the cryoprotective activity of these peptides toward GST (Fig. 2B) and the results were similar to those found with LDH. Also, D10[20] was again the shortest peptide with reasonably sufficient cryoprotective activity toward GST at 50 μg/ml. We further analyzed the normalized cryoprotective activity by amino acid lengths and calculated isoelectric points, pI (Figs. 2C and 2D, respectively). In order to compare the cryoprotective activities for the different reporter enzymes, both the maximum preserved enzymatic activities with the cryoprotective peptide, LDH with E1[31] and GST with D10[20], were set to 100%, respectively. Peptides shorter than 20 amino acids were less cryoprotective, which is consistent with the proposed cryoprotection mechanism of DHNs as a molecular shield effect, where cryoprotective activity of various DHNs were roughly proportional to $R_H$ logarithms of the cryoprotectants (Cuevas-Velazquez, Rendon-Luna & Covarrubias, 2014; Ferreira et al., 2018). Since we only examined cryoprotective IDPs smaller than 42 residues, only the lower limit of the correlation has been observed. In addition, we found that the IDP peptides with either extremely high or low pI were less cryoprotective, whereas the peptides with neutral pI were more potent.

Hereafter, our shortest cryoprotective IDP-derived peptide D10[20] is referred to as FK20.

## Cryoprotective activity of FK20 and its all-D-enantiomeric isomer FK20-D

We further characterized cryoprotective activity and its FK20 mechanism by using its all-D-enantiomeric isomer, FK20-D. Based on this study and our previous study, the cryoprotective activity of IDP-derived peptides is likely not amino acid sequence specific. For example, we measured and compared $^1H$-$^{15}N$ HSQC spectra of the target molecule Aβ(1-42) in the absence and the presence of the cryoprotective IDPs in our previous study, and we observed almost no spectra change (Ikeda et al., 2020). Although it is difficult to completely prove the absence of any specific interaction between the IDPs and the reporter enzymes, we challenged to provide indirect evidence to support this assumption as many as possible. In this study, we examined the cryoprotective activity of FK20-D and compared to that of the parent FK20. The cryoprotective activities of FK20, FK20-D, and the same total concentration of the racemic equimolar mixture of FK20 and FK20-D (FK20-LD) toward LDH and GST at varying concentrations were examined (Figs. 3A and 3B, respectively). FK20-D showed similar concentration-dependent cryoprotective profiles toward the reporter enzymes. The results suggested that specific molecular interaction between the reporter enzymes and either the FK20 or FK20-D peptides was absent, because the contribution of any residue-specific interaction of FK20 is not reproducible with FK20-D. We assumed that the effect of FK20 and FK20-D toward LDH and GST were more "environmental". Indeed, FK20-LD showed similar concentration-dependent cryoprotective profiles (Figs. 3A and 3B). Note that the additive property of different cryoprotective IDPs was also observed between full length C1 and D10 (Fig. S2). A 1:1 mixture (in weight) of C1 and D10 showed a concentration-dependent cryoprotective profile similar to that of C1.

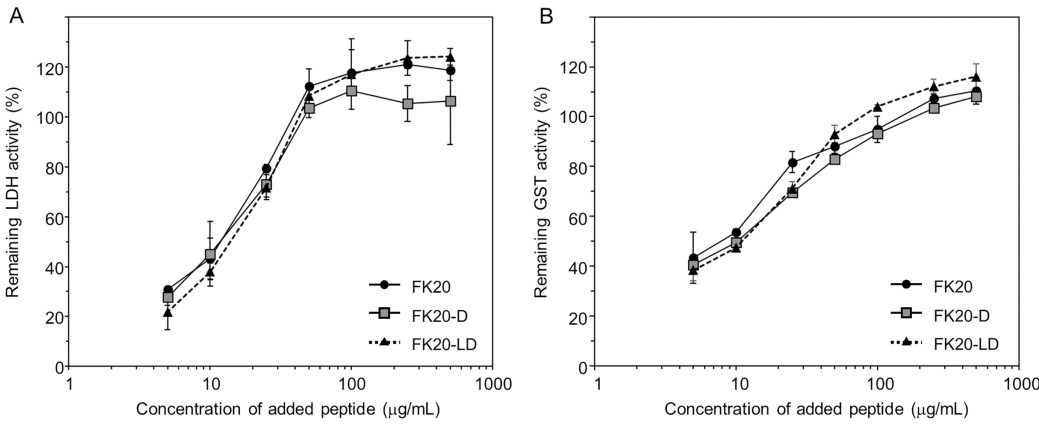

**Figure 3 Cryoprotective activity of IDP-derived peptide FK20 and its all D-enantiomeric isomer FK20-D towards LDH and GST.** Cryoprotective activity of IDP-derived peptide FK20 and its all D-enantiomeric isomer FK20-D towards LDH (A) and GST (B) (A) LDH activities were plotted (final concentration of 50 μg/ml) after freeze-thawing in the presence of additive IDP peptides of concentrations ranging from 2.5 to 500 μg/ml. The LDH activity of the untreated sample was set to 100%. FK20: filled circle, FK20-D: grey square. (B) GST activities were plotted (final concentration of 280 μg/ml) after freeze-thawing in the presence of additive IDP peptides of the concentration ranging 2.5 to 500 μg/ml. The GST activity of the untreated sample was set to 100%. FK20: filled circle, FK20-D; grey square, FK20-DL (equimolar mixture of FK20 and FK20-D): filled triangle with dashed line.

## Effect of cryoprotective IDP peptide toward structure and thermal stability of GST and LDH

Next we tried to show that the cryoprotective activity was *not* due to either structural change or thermal stabilization of GST, but rather that reporter enzyme cryoprotection can be explained by avoiding denaturation of the enzymes by structural or thermal stabilization through either specific or non-specific IDP interactions. In our the other previous study, we succeeded in demonstrating the amyloid formation inhibitory activity of the same human genome-derived IDP peptides against Aβ(1-42) (*Ikeda et al., 2020*). At that time, we employed solution NMR techniques to monitor existence of a specific molecular interaction between $^{15}$N-labelled Aβ(1-42) in the presence of non-labelled IDPs. However, in this study, the molecular weight of the reporter enzyme (for example, GST) seems not suitable for solution NMR. Therefore, we employed CD spectroscopy, since FK20-LD, the equimolar mixture of FK20 and FK20-D, is silent in CD measurement. Figure 4A presents the experimental evidence of this unique feature of FK20-LD (grey), showing no significant CD band. FK-20 (black) showed the typical CD spectrum of a disordered state, whereas FK20-D (black, dashed line) showed the exact same CD spectrum with an inverted sign, as expected.

Figure 4B shows the CD spectra of GST in the absence (dashed line) and presence (solid line) of FK20-LD. There was no significant change in CD spectra, indicating that GST did not change its three-dimensional structure. Thus, FK20-LD cryoprotective activity was not a result of GST structural change. Figures 4C and 4E show a series of GST CD spectra at increasing temperatures from 25 °C to 75 °C with and without FK20-LD. Figure 4D and 4F show GST thermal denaturation plots taken from Figs. 4C and 4E,

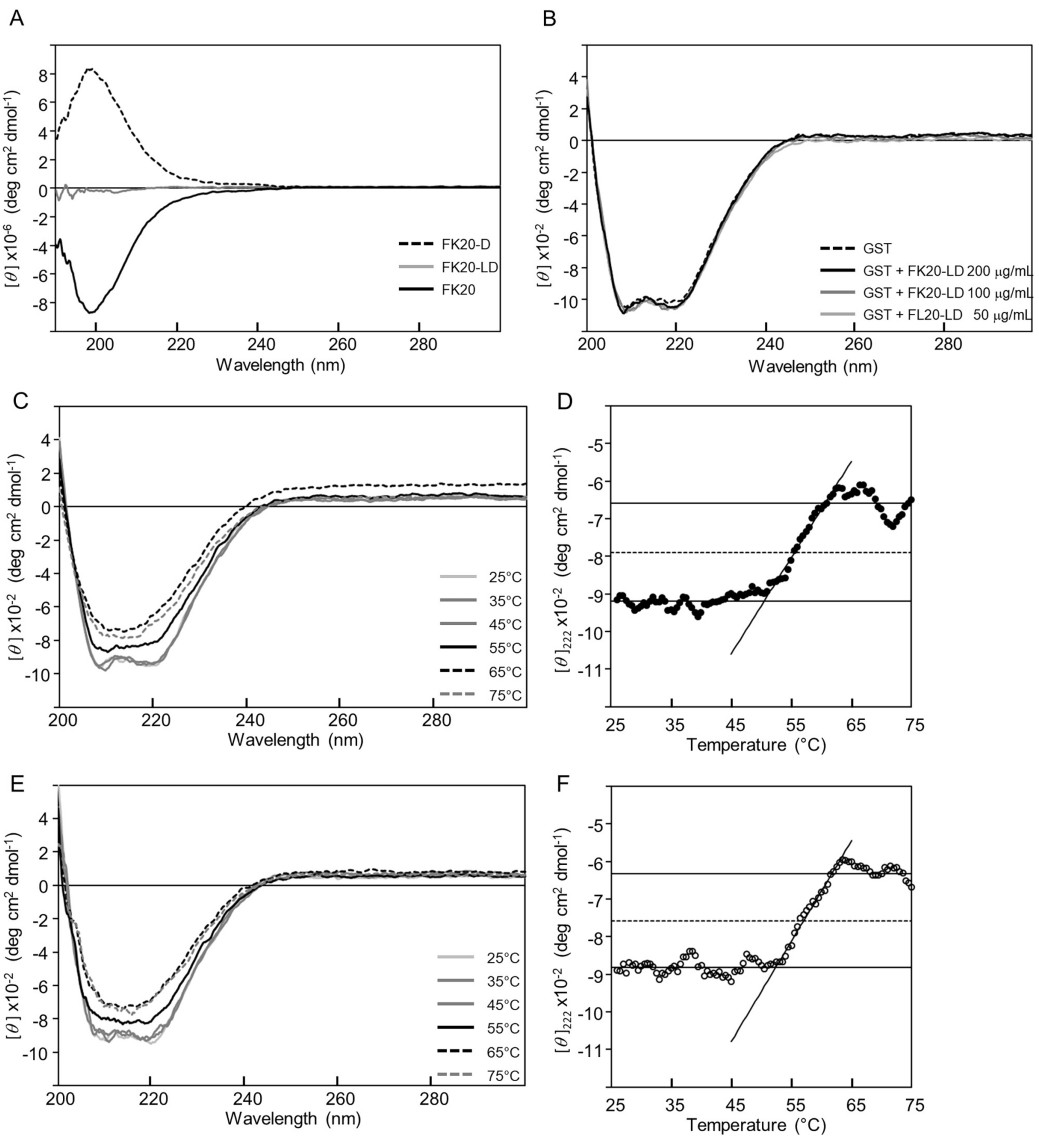

**Figure 4 Influence of IDP-derived peptide FK20 and its all D-enantiomeric isomer FK20-D to the structure and thermal stability of GST.** Influence of IDP-derived peptide FK20 and its all D-enantiomeric isomer FK20-D to the structure and thermal stability of GST. (A) CD spectra of 250 μg/ml of FK20 (solid line), FK20-D (dashed line) and equimolar mixture (125 μg/ml each) FK20-LD (grey line) at 25 °C. (B) CD spectra of 100 μg/ml GST without (dashed line) or with (solid lines) 50, 100, and 200 μg/ml of FK20-LD at 25 °C. (C) CD spectra of 100 μg/ml GST alone at various temperatures. (E) CD spectra of 100 μg/ml GST with 200 μg/ml of FK20-LD at various temperatures ranging 25 °C to 75 °C. (D, F) Melting curve of GST with (F) or without (D) FK20-LD monitored by the residual molar ellipticity [theta] at 222 nm.

respectively. The GST thermal denaturation temperature ($T_m$) was 56.5 ± 1.2 °C for GST alone, and 58.2 ± 1.1 °C in the presence of 0.2 mg/ml of FK20-LD. Similarly, Fig. 5A shows the CD spectra of LDH in the absence (dashed line) and presence (solid line) of FK20-LD. Figures 5B and 5D show initial and final LDH CD spectra at increasing temperatures from 25 °C to 75 °C with and without FK20-LD. Figures 5C and 5E show LDH thermal denaturation plots taken from Figs. 5B and 5D s, respectively. The LDH

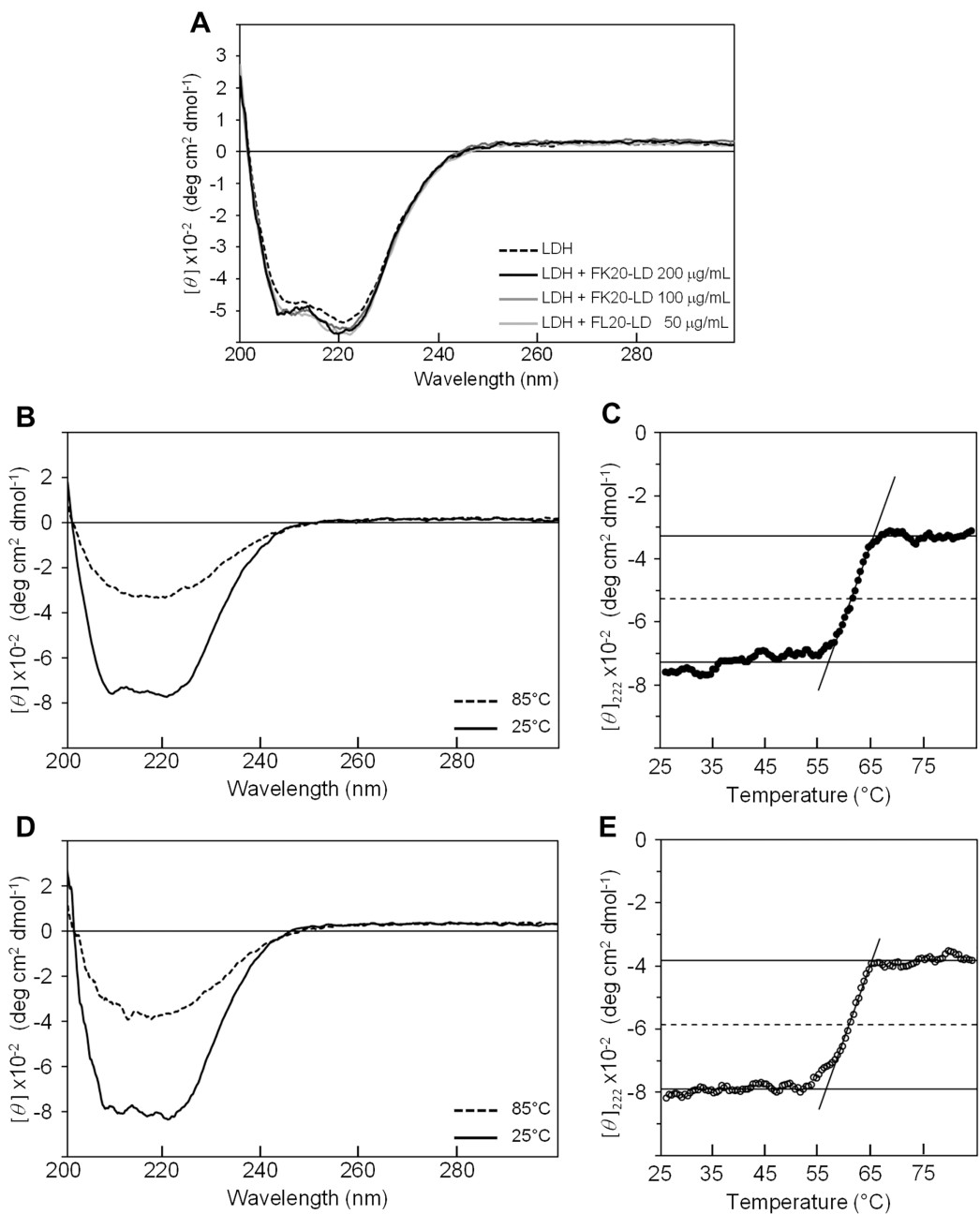

**Figure 5 Influence of IDP-derived peptide FK20 and its all D-enantiomeric isomer FK20-D to the structure and thermal stability of LDH.** Influence of IDP-derived peptide FK20 and its all D-enantiomeric isomer FK20-D to the structure and thermal stability of LDH. (A) CD spectra of 100 µg/ml LDH without (dashed line) or with (solid lines) 50, 100, and 200 µg/ml of FK20-LD at 25 °C. (B) CD spectra of 100 µg/ml LDH alone at two temperatures, 25 °C and 85 °C. (D) CD spectra of 100 µg/mL LDH with 200 µg/ml of FK20-LD at two temperatures, 25 °C and 85 °C. (C, F) Melting curve of LDH with (E) or without (C) FK20-LD monitored by the residual molar ellipticity (theta) at 222 nm.

thermal denaturation temperature ($T_m$) was 61.2 °C for LDH alone, and 62.0 °C in the presence of 0.2 mg/ml of FK20-LD. The results suggest that the thermal stability gains of GST and LDH by FK20-LD are small. Thus, we concluded that the FK20-LD

cryoprotective activity was likely not a result of a thermal stabilization effect on the reporter enzymes.

## DISCUSSION

In this study, we succeeded in minimizing the amino acid length of human-genome-derived IDPs with a reasonable cryoprotective effect into 20 amino acid residues. The shortest cryoprotective peptide, FK20, showed 100% cryoprotective action toward LDH at a concentration of 100 µg/ml (0.01%). We observed that the peptides shorter than 20 amino acids were less cryoprotective. The result is consistent with the proposed cryoprotection mechanism of DHNs as a molecular shield effect, where cryoprotective activity of various DHNs were roughly proportional to hydrodynamic radius $R_H$ logarithms of the cryoprotectants (*Cuevas-Velazquez, Rendon-Luna & Covarrubias, 2014*; *Ferreira et al., 2018*). The mechanism of molecular shield is to inhibit a core of protein aggregation during freeze-thaw cycles of the proteins by disturbing the direct collision between the reporter enzymes. Thus, $R_H$ of cryoprotectant is considered for explaining the molecular shield efficacy.

We demonstrated that the cryoprotective activity of FK20 is independent to its molecular chirality, that suggests the absence of the specific interaction between the reporter enzymes and FK20. This assumption is consistent with our observation that the addition of IDP (in this time, FK20/FK20-D mixture) did not affect the thermal stability of GST and LDH. The GST thermal denaturation temperature ($T_m$) was 56.6 ± 1.1 °C for GST alone, and 58.1 ± 1.1 °C in the presence of 0.2 mg/ml of FK20-DL (Figs. 4D and 4F). Similarly, the $T_m$ of LDH was 61.2 °C for LDH alone, and 62.0 °C in the presence of 0.2 mg/ml of FK20-DL, and the remarkable gain of thermal stability was not observed (Figs. 5C and 5E). Thus, we concluded that the FK20-DL cryoprotective activity was likely not a result of a thermal stabilization effect on the reporter enzymes.

The FK20 mechanism of cryoprotective activity toward the reporter enzymes seems to be a molecular shield effect, in which the peptide may prevent stochastic contacts with the reporter enzyme molecules upon desiccation during the freezing process. The schematic diagrams of cryoprotecting mechanisms toward the reporter enzymes are summarized in Fig. 6. Recently, a new class of functional IDPs that exhibits a potent protective activity against protein aggregation following to heat-denaturation, named heat-resistant obscure (Hero) proteins was discovered (*Tsuboyama et al., 2020*). Although the mechanism of aggregation protection of Hero proteins is not yet unraveled, the molecular shield mechanism is one of the candidates. We recently reported that human genome-derived IDPs also can inhibit the nucleation phase of the amyloid fibril formation of Aβ(1-42) (*Ikeda et al., 2020*), and the fibril formation inhibition is likely based on the molecular shield mechanism.

In conclusion, we have developed a cryoprotective peptide, FK20, with 20 amino acid residues. The sequence of FK20 was taken from human genome encoded intrinsically disordered proteins. The mechanism of cryoprotection toward the reporter enzymes, LDH and GST, was assumed to be a molecular shield effect with no specific molecular

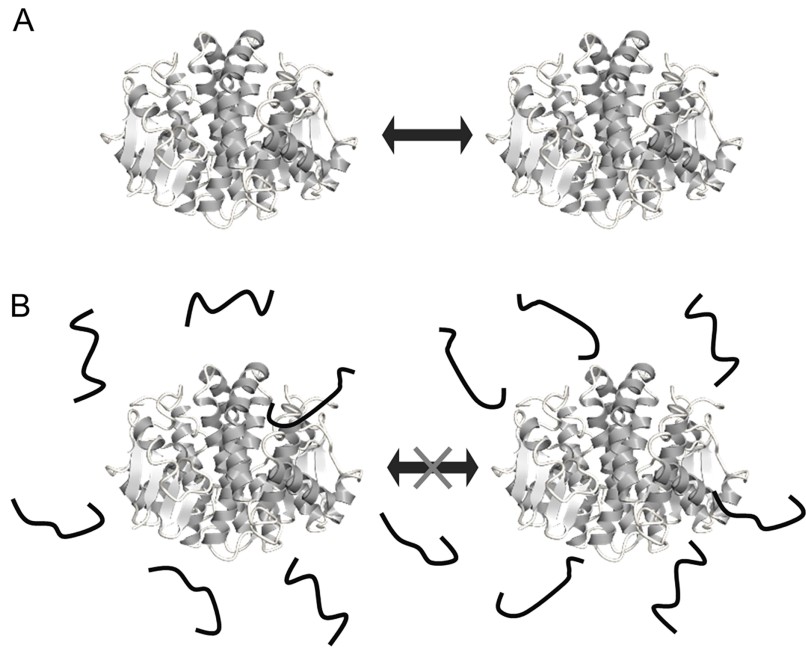

**Figure 6 Schematic representation of the mechanisms for cryoprotection of enzyme by FK20.** Schematic representation of the mechanisms for cryoprotection of enzyme by FK20. (A) Illustration of the mechanism of stochastic protein collisions upon repeated freeze/thaw cycles. The GST molecule (dimer) is represented by the ribbon diagram. The filled block arrow indicates stochastic direct collision of proteins during freeze/thaw cycles. (B) Illustration of the molecular shield model of FK20 cryoprotective activity toward the enzymes, originally proposed by Chakrabortee et al. (see text) (*Chakrabortee et al., 2012*). FK20 is represented as bold strings.

interaction. Taking into consideration of these facts, the molecular shield becomes one of noteworthy properties of IDPs.

## CONCLUSIONS

We have developed a cryoprotective peptide, FK20, with 20 amino acid residues. The sequence of FK20 was taken from human genome encoded intrinsically disordered proteins. FK20, its all-D-enantiomeric isomer FK20-D, and the equimolar mixture of FK20/FK20-D exhibited the similar level of cryoprotective activity toward the reporter enzymes, LDH and GST. No marked thermal stabilization of GST was observed in the presence of the equimolar mixture of FK20/FK20-D. The mechanism of cryoprotection toward the reporter enzymes, LDH and GST, was assumed to be a molecular shield effect with no (stereo-)specific molecular interaction.

### Funding

This work was supported by JSPS KAKENHI Grant Number 21113007, the Salt Science Research Foundation Grant Number 1222, and AMED Project Focused on Developing Key Evaluation Technology, Grant Number 16be0204437h0003. The funders had no role

in study design, data collection and analysis, decision to publish, or preparation of the manuscript.

## Grant Disclosures
The following grant information was disclosed by the authors:
JSPS KAKENHI: 21113007.
Salt Science Research Foundation: 1222.
AMED Project: 16be0204437h0003.

## Competing Interests
Hidekazu Hiroaki and Takeshi Tenno are the founders of BeCellBar LLC*., a Nagoya University origin start-up company of biotechnology. There are no competing interest for the other authors.

## Author Contributions
- Naoki Matsuo performed the experiments, analyzed the data, prepared figures and/or tables, and approved the final draft.
- Natsuko Goda performed the experiments, analyzed the data, prepared figures and/or tables, and approved the final draft.
- Takeshi Tenno performed the experiments, prepared figures and/or tables, and approved the final draft.
- Hidekazu Hiroaki conceived and designed the experiments, performed the computation work, authored or reviewed drafts of the paper, and approved the final draft.

## Data Availability
The raw data is available in the Supplemental Files.

## Supplemental Information
Supplemental information for this article can be found online at http://dx.doi.org/10.7717/peerj-pchem.20#supplemental-information.

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
