# Peer review of "Cryoprotective activities of FK20, a human genome-derived intrinsically disordered peptide against cryosensitive enzymes without a stereospecific molecular interaction"

_PeerJ Physical Chemistry, doi:10.7717/peerj-pchem.20_

## Round 0.1 · original submission · Major Revisions

Hello. The reviewers and I find the work interesting and significant. There are serious concerns about making the paper more readable and organized. I hope this can be addressed by thorough editing. Please also address the concerns of the reviewers about the figures.

Reviewer 1 ·

Basic reporting

This manuscript reports cryoprotective activities of three human-genome derived IDPs and their truncated peptides for LDH and GST. CD analysis by using D-enantiomeric isoform suggested that FK20 might not interact to GST and not affect the GST's structure during the cryoprotection. FK20 could also successfully cryopreserve two mammalian cell systems.

Experimental design

The experimental design is modern and appropriate.

Validity of the findings

I have major difficulties to understand this finding. Details are shown in the following "General comments."

Additional comments

1) It is my doubt thorough the manuscript that the information of the gene (protein) is limited. The gene information supports readers to correctly understand this work. First, it is needed to note the reason why authors used these protein sequences. More than 10% of open reading frames in eukaryotic genomes have been estimated as intrinsically disordered proteins or proteins containing intrinsically disordered regions. Authors should show the intension of choosing the gene from abundant IDPs existing in the human genome in the "Introduction" section or the "Results" section. Second, it is highly recommended that the accession number, the function (unless it has determined, putative), and the expression patterns (organ, age, stress specificities) of the protein are documented in the text.

2) Figure 1. This figure is difficult to understand. What does the green, blue, and magenta bars represent? Although some bars have amino acid sequences above them, they did not fit to the lengths of the corresponding color bars (or each amino acid sequence is a part of the original sequence that was exhibited by a bar?). Are these sequences parts of some proteins? if so, show which part is taken from the original protein.

3) Figures 2, 3, and 5. Please show basic information in the legends to understand the figures. How many times the experiments are repeated? What is the mean of error bars? Statistically significant?

4) Figure 6. A) the paper (Chakrabortee et al. 2012) proposed two types of molecular shield models, original one and extended one. Figure 6A seems to show the extended one. However, applying extended model needs weak association between cryoprotective peptides and protected proteins. Authors should describe the reason why they adopted the extended molecular shield model to their results. B) I think that relevance of the data is not enough to make Figure 6B.

5) The title is somewhat ambiguous. A title which directly represents the cryoprotective activities of FK20 for cryosensitive enzymes and mammalian cells seems to be better.

6) L45-46. The phrase "but also non-functional proteins" can be omitted because the following sentence mentioned "IDPs are widely associated with biological processes".

Reviewer 2 ·

Basic reporting

The manuscript contains numerous small grammatical errors that needs correction by a native English speaker. Below are corrections for text that are not necessarily grammatically incorrect:

Line 29 Change “if all-D-enantiomeric” to “of all-D-enantiomeric”
Line 34 Change “IDPs larger than 20 amino acids” to “IDPs longer than 20 amino acids”
Line 46 What is meant by “non-functional” proteins? Is this in the sense of not yet identified, or that they have zero biological function?
Line 49 Not all IDPs will fold into a stable structure, or gain any structure, when bound to a ligand. Please reword this sentence to reflect that.
Line 91-104 This section could be shortened since it reads like a very detailed summary of the results. I would also suggest using the present tense to avoid confusion.
Line 213 “It is generally saying that proving non-existence is always difficult.” To be clear, it is impossible to prove something does not exist because we cannot distinguish between something not existing or that we cannot observe it. Please rewrite this sentence.
Line 222 “atmospheric” - perhaps the author means environmental?
Line 216+ I assume the equimolar mixture concentration is the total concentration of the protein?

There are several problems with the figures. Fig. 1 is very confusing to me - the protein sequence text is not the same length as any of the coloured bars, making it difficult for me to tell which sequence corresponds to what bar. PONDR is mentioned in the text on line 185, but I could not find any such plot (Fig. S1 is actually a Uversky plot). The supplemental filenames should be revised to make it clearer which files correspond to Fig. S1 etc.

Experimental design

No comment

Validity of the findings

The paper has a considerable amount of interesting data and the use of all-D amino acids is a very clever way to address the molecular shield mechanism of IDPs, and an especially smart way to eliminate the IDP signal in the GST CD studies.

I am having some difficulty interpreting Fig. 2C and 2D. Normalized activity is not defined in the Methods and I do not see the calculation in the Excel files. Because of the sigmoidal shape of the LDH cryoprotection curves, using a single protein concentration does not provide an accurate view of the efficiency. I would strongly suggest plotting the normalized activity as PD50 (concentration of protein required to recover 50% of LDH activity) to get a hopefully better view of how these IDPs are working.

The cell cryoprotection data are interesting, but I think they are very out of context in this manuscript. I think the data should be removed, because at this point it is not clear if the protection is due to the protein stabilization effect seen in the in vitro experiments, or due to membrane stabilization. The membrane stabilization is a possibility, because the IDPs were added to the culture media and there is no specific evidence that these peptides are internalized.

---

## Round 0.2 · accepted · Accept

I thank the authors for their strong efforts to address the reviewers' criticisms. This work is intriguing and is acceptable for publication.

Reviewer 2 ·

Basic reporting

No comment

Experimental design

No comment

Validity of the findings

No comment

Additional comments

The authors have made all of the appropriate changes based on my review.